# Isolation and Identification of *Acer truncatum* Endophytic Fungus *Talaromyces verruculosus* and Evaluation of Its Effects on Insoluble Phosphorus Absorption Capacity and Growth of Cucumber Seedlings

**DOI:** 10.3390/jof10020136

**Published:** 2024-02-08

**Authors:** Qingpan Zeng, Jiawei Dong, Xiaoru Lin, Xiaofu Zhou, Hongwei Xu

**Affiliations:** Jilin Provincial Key Laboratory of Plant Resource Science and Green Production, Jilin Normal University, Siping 136000, China; 13474205037@163.com (Q.Z.); 15043441213@163.com (J.D.); lxr18243521165@163.com (X.L.); xuhongwei@jlnu.edu.cn (H.X.)

**Keywords:** endophytic fungi, phosphorus solubilization, plant growth promotion

## Abstract

The symbiosis between endophytic fungi and plants can promote the absorption of potassium, nitrogen, phosphorus, and other nutrients by plants. Phosphorus is one of the indispensable nutrient elements for plant growth and development. However, the content of available phosphorus in soil is very low, which limits the growth of plants. Phosphorus-soluble microorganisms can improve the utilization rate of insoluble phosphorus. In this study, *Talaromyces verruculosus* (*T. verruculosus*), a potential phosphorus-soluble fungus, was isolated from *Acer truncatum*, a plant with strong stress resistance, and its phosphorus-soluble ability in relation to cucumber seedlings under different treatment conditions was determined. In addition, the morphological, physiological, and biochemical indexes of the cucumber seedlings were assessed. The results show that *T. verruculosus* could solubilize tricalcium phosphate (TCP) and lecithin, and the solubilization effect of lecithin was higher than that of TCP. After the application of *T. verruclosus*, the leaf photosynthetic index increased significantly. The photosynthetic system damage caused by low phosphorus stress was alleviated, and the root morphological indexes of cucumber seedlings were increased. The plant height, stem diameter, and leaf area of cucumber seedlings treated with *T. verruculosus* were also significantly higher than those without treatment. Therefore, it was shown that *T. verruculosus* is a beneficial endophytic fungus that can promote plant growth and improve plant stress resistance. This study will provide a useful reference for further research on endophytic fungi to promote growth and improve plant stress resistance.

## 1. Introduction

Pi is an essential element and plays a crucial role in plant growth [1,2]. Pi is the main component of biofilm and ATP in plants, which play an intermediary role in the process of nutrient absorption and information transfer [3]. ATP can mediate the signal transduction network to activate the energy metabolism pathway. Pi is also a substrate of the enzyme-catalyzed reaction, which participates in photosynthesis and respiration, and regulates enzyme activity to ensure normal plant growth and development [4,5]. The photosynthetic rate of sugar beet decreased significantly due to a lack of phosphorus [6]. However, phosphorus in soil reacts easily with minerals to form insoluble compounds [7], which cannot be directly absorbed and utilized by plants, thus limiting plant growth [8,9]. Pi is a nonrenewable resource [10]. At present, there is an insufficient soil phosphorus supply globally, and the lack of phosphorus directly affects crop yields [11,12]. Developing phosphorus-soluble microorganisms is an environmentally friendly and economical solution [13,14]. Therefore, it is particularly important to search for microorganisms that increase soil available phosphorus.

*Acer truncatum* is a small deciduous tree of the Aceraceae family that is widely distributed in northern China [15]. It has certain ornamental and medicinal value [16,17,18]. *Acer Truncatum* can still grow normally in soil with low phosphorus content and has certain stress resistance [19], which may be related to the abundance of endophytic fungi in its body. Endophytic fungi are widely distributed in plants and form a mutualistic relationship with their hosts. In the process of long-term coevolution, endophytes develop an increasingly close relationship with host plants. On the one hand, they obtain substances needed for their own growth from plants, and on the other hand, they provide abundant secondary metabolites for plants [20,21]. These products help plants resist pests and diseases, drought, heavy metals, and extreme environmental stress, thus promoting plant growth. For example, endophytic fungi in mangroves in marine swamps can improve host salt tolerance and alkaline tolerance [22], and endophytic fungi in desert plants can improve host tolerance to high temperatures [23]. Therefore, it was feasible to isolate endophytic fungi with phosphorus-soluble properties from *Acer truncatum* that could tolerate low phosphorus stress.

Phosphorus dissolution is a key way for microorganisms to promote plant growth [24]. According to Alori et al., different types of organic acids produced by microorganisms cause P ions to be replaced by cations, converting insoluble phosphorus into soluble phosphorus forms in plants and increasing the phosphorus content [13]. In adversity, how to effectively enhance photosynthesis is of great significance to increase crop yield. Therefore, how to improve the photosynthetic capacity of crops deserves attention [25]. Studies have shown that the endophytic fungi Trichoderma viride can increase soil available phosphorus content, increase phosphorus absorption by Melilotus officinalis, improve plant root morphology, and regulate plant photosynthesis to promote plant growth [26]. However, phosphorus-solubilizing microorganisms may compete with plants for nutrients in environments with low Pi content [27], so more experiments need to be conducted to screen and verify the application of phosphorus-solubilizing microorganisms to improve crop yield [26]. At present, most phosphorus-soluble strains were isolated from plant rhizosphere soil [28,29,30]. The aim of this study was to isolate endophytes with phosphorus-soluble properties from *Acer truncatum*, to provide a theoretical basis for further understanding the mechanism of *Acer truncatum*’s stress resistance, and to enrich the phosphorus-soluble microbial resources [31]. In addition, the effects of *T. verruculosus* inoculation on the physiological characteristics of cucumber seedlings were also studied to verify whether *T. verruculosus* had the effect of promoting plant growth, which provides a new basis for the further preparation of phosphorus-soluble microbial bactericides and promotes the sustainable development of agriculture.

## 2. Materials and Methods

### 2.1. Experimental Materials

The *Acer truncatum* used in the extraction of endophytic fungi was derived from Tiexi District, Siping City, Jilin Province (altitude 150 m, 124°20′30.84″ E, 43°9′37.8″ N). The isolated *T. verruculosus* was sent to the General Microbiology Center of China Microbiological Culture Preservation and Management Committee (CGMCC; No. 40727) for storage [26]. It was named Y-BC-JYLZJ. The test medium was potato glucose AGAR (PDA), the test plant was cucumber (Jinyan No. 4), and the soil was taken from the pear tree experimental field in Siping, Jilin Province.

### 2.2. Isolation and Purification of Endophytic Fungi from Acer truncatum

In this study, three stems of the same length were selected from three *Acer truncatum* plants.

The stem part of *Acer truncatum* was cut into 1 cm sections and washed with running water for 20 min. The surface moisture was removed with absorbent paper. It was soaked in 2% sodium hypochlorite for 15 min. Then, it was washed with sterile water 3 times, disinfected with 75% ethanol for 1 min, washed with sterile water 3 times, and the water on the surface was absorbed using sterile filter paper. After the aseptic filter paper had dried the water, the *Acer truncatum* stem segment was placed flat on the PDA medium for culturing. Then, the PDA medium with the *Acer truncatum* stem segment was placed in the 28 °C incubator for 3d to observe whether colony formation occurred. After the formation of a single colony, the strains with different forms and colors were immediately transferred onto a new PDA medium for culture purification and preservation of the strains.

### 2.3. Identification of Endophytic Fungi from Acer truncatum

Fungal genomic DNA was extracted by the SDS method, and the extracted genomic DNA was amplified by PCR. The PCR amplification system was 25 μL, including 9.5 μL ddH_2_O, 2 μL primers (ITS-1, ITS-4), and 12.5 μL 2 × Taq PCR Master Mix. In total, 1 μL template DNA was amplified with the fungal universal primer ITS-1(5′-TCCGTAGGTGAACCTGCGG-3′)ITS-4(5′-TCCTCCGCTTATTGATATGC-3′), and the reaction conditions were as follows: predenaturation at 94 °C, 5 min; denaturation 94 °C, 30 s; annealing 55 °C, 30 s; extension 72 °C, 50 s; 30 reaction cycles; extension 72 °C for 5 min [32]. The PCR products were identified by 1% agarose gel electrophoresis, and the PCR products were sent to Beijing Liuhe BGI Gene Technology Co., Ltd. (Beijing, China) for sequencing. The sequencing results were compared with the BLAST algorithm and the National Center for Biotechnology Information (NCBI) database.

### 2.4. T. verruculosus Phosphorus Solubilization Pi Experiment

For the preparation of pH7.0 NBRIP (National Botanical Research Institute’s Phosphate) culture solution [33], the culture medium was treated with Ca_3_(PO_4_)_2_, Ca_3_(PO_4_)_2_ + *T*. *verruculosus* (T), Lecithin, and Lecithin + T with 3 repeats per treatment. The concentration of spore liquid was 1 × 10^8^ cfu/mL. After shock culture at 28 °C and 166 rmp for 10 days, the Pi concentration of culture filtrates and mycelium was detected.

### 2.5. Potted Plant Experiment

In order to accurately study the effect of *T. verruculosus* in mediating plant absorption of phosphorus, this study selected cucumber as the experimental material for a total of eight treatments, as follows: normal concentration Pi: 1 mM KH_2_PO_4_ (P); Pi+ *T. verruculosus* (PT); apply low concentration Pi:10 μm KH_2_PO_4_ (DP); 1%Pi+ T (DPT); 1%Pi +Ca_3_(PO_4_)_2_ (TP); 1%Pi + Ca_3_(PO_4_)_2_ +T (T%Pi + T); 1%Pi + Lecithin (LP); 1%Pi + lecithin + T (LP + T). Each treatment contained 3 pots, and each pot contained two seedlings. The flowerpots were 20 cm in diameter and 17 cm in height. The experimental treatment started with four leaves of one heart stage. The spore solution was diluted to 1 × 10^8^ cfu/mL, and then 10 mL of spore solution was injected into the root of each seedling by syringe once a week for a total of 4 weeks. Hoagland nutrient solution 100 mL with a phosphorus deficiency of 1/2 was poured once a week into each basin. The soluble phosphorus in the applied phosphorus was KH_2_PO_4_, the normal phosphorus concentration was 1 mmol/L, the low phosphorus concentration was 10 μmol/L, the inorganic insoluble phosphorus source was Ca_3_(PO_4_)_2_, and the organic insoluble phosphorus source was yolk lecithin. The K + concentration in the nutrient solution was supplemented with KCI. Cucumber seedlings were harvested after four weeks of treatment, and a physiological and biochemical analysis was performed (Figure 1). During the growth of cucumber seedlings, the temperature was 28 °C during the day, 18 °C at night, the humidity was 60~70%, and the light conditions were 14 h light and 10 h darkness.

### 2.6. Determination of Morphological Index

The height of the plant was measured with a ruler, and the diameter of the stem was measured with a vernier caliper. After sampling, the leaves were placed on A4 paper and the ruler was added as a control. Images were obtained by digital photography, and the area of the leaves was calculated by the Image J-pro 6.0 software. The roots of plants were sampled. Six plants of similar growth were selected for each treatment. The roots of the cucumber plants were scanned with an Epson 11000 scanner (Epson America, Inc., Long Beach, CA, USA). The analysis was carried out with the Win RHIZO 2012b (Regent, Vancouver, BC, Canada) root analysis system, and the obtained images were further analyzed for root length, root surface area, root volume, and other characteristics. The cucumber plants were washed with clean water, and the fresh samples of the aboveground part and the underground part were placed in an oven at 105 °C for 30 min, and dried at 75 °C to constant weight. The dry weight was determined by an electronic balance.

### 2.7. Determination of Pi Content in Cucumber Plants

We used the H_2_SO_4_-H_2_O_2_ digestion-molybdenum-antimony resistance colorimetric method for the determination of the Pi content of cucumber plants [34].

### 2.8. Determination of Nutrient Content in Soil

The soil was collected and mixed well. Then, 5 g of air-dried soil was weighed with a 2 mm screen and put into 50 mL beakers, before 12.5 mL of 0.01 mol·L^−1^ CaCl_2_ solution was added. Thereafter, it was shaken in a 180 r min^−1^ shaker for 1 h. After standing for 30 min, the pH of the soil was determined with a pH meter. The contents of ammonium, nitrogen, available phosphorus, available potassium, and organic matter in soil were determined by the colorimetric method with a high-precision soil fertilizer nutrient detector (TY-04+, Zhengzhou Tengyu Instrument Co., Ltd., Zhengzhou, China).

### 2.9. Determination of Photosynthetic Pigment Content

For each treatment, 1.0 g of leaves was soaked in 10 mL of 95% ethanol until the leaves turned white, and the extracts were centrifuged at 10,000 rpm for 10 min (the supernatant was taken). The absorbance at 663 nm, 645 nm, and 652 nm was measured by a spectrophotometer [26]. The calculation formula is as follows: Chl a = (12.21 × A_663_) − (2.81 × A_645_), Chl b = (20.13 × A_645_) − (5.03 × A_663_), Chl a/b = Chl a/Chl b, *p* (mg L^−1^) = 20.29A_645_ + 8.05A_663_.

### 2.10. Chlorophyll Fluorescence Parameters

Chlorophyll fluorescence parameters were determined using an imaging-PAM-modulated chlorophyll fluorescence IMAGING system (MINI-IMAGING-PAM, Heinz Walz, Effeltrich, Germany). The leaves were dark-adapted for 30 min before measurement. The maximum quantum yield (Fv/Fm), photochemical quenching coefficient (qP), nonphotochemical quenching coefficient (NPQ), and effective quantum yield of PSII in photosystem II (PSII) were determined, and a line diagram of fluorescence parameters varying with light intensity was drawn [35]. Through the measurement of a 16-step fast light curve (rlc), the line graph of fluorescence parameters with light intensity was drawn. Table 1 lists the parameters and calculation formulas.

### 2.11. Transient Measurement of OJIP and Kinetic Analysis of Rapid Fluorescence Induction

The instrument used was a Handy-PEA (Handy plant efficiency analyzer, Hansatech Instruments Ltd., Pentney, UK), and the leaf clip was placed on the front of the leaf for 20 min. Six detection sites were selected for each blade, and three detection points were selected for each blade along both sides of the main vein, with the four points being equally spaced from the tip of the blade to the base of the blade. After dark adaptation, the instrument probe was connected with the blade clamp on the blade, and the slide switch was opened to expose the measurement hole to the laser light source. Using the preset LED light source of 3000 μmol·m^−2^·s^−1^ and the detection time of 1 s, rapid fluorescence signal acquisition was carried out. Finally, the fluorescence signals of 6 detection points on each leaf were averaged and used as the final fast fluorescence data of the sample [36]. The normalization formula for order O and order P is W_O-P_ = [(F_t_ − F_o_)/(F_m_ − F_o_)]. The normalization formula for orders O and J is W_O-J_ = [(F_t_ − F_o_)/(F_J_ − F_o_)]. The extraction parameters and formulas of chlorophyll (Chl) a fluorescence (OJIP) transient data are shown in Table 2. 

### 2.12. Statistical Analysis

IBM SPSS 22.0 was used for the statistical analysis in the experiment [37]. All data are expressed in the form of the mean ± standard deviation of 6 repeats, and one-way ANOVA was used to analyze the significance of differences between *T. verruculosus* and *T. verruculosus,* without *T. verruculosus,* and between different phosphorus forms (*p* < 0.05). Charting was conducted with OriginPro 2022 (OriginLab, Northampton, MA, USA).

## 3. Results

### 3.1. Results of Blast Sequence Comparison of Endophytic Fungi of Acer Truncatum

In this study, the genomic DNA of 21 strains of fungi was used as the material for the molecular detection of each strain, and the 16S rRNA gene sequencing results of 21 strains of fungi were compared in the NCBI database. After sequencing and comparison, 10 strains were obtained (Table 3). They fall into six genera: *Basiliformis*, *Trichoderma*, *Trichoderma*, *Mucor*, *Streptispora,* and *Parasitospora*. According to relevant reports, *Talaromyces flavus* has the ability to dissolve phosphorus [38]. In addition, *Talaromyces flavus* and strain 19 are from the same genus of fungi; thus, strain 19 was further studied, and the phylogenetic tree of strain 19 Y-BC-JYLZJ was constructed using the MEGA 7.0 software [39]. The results showed that Y-BC-JYLZJ was most similar to MW081280.1 *T. verruculosus* strain 2-F1, with a similarity of 100% (Figure 2). Combined with the morphological characteristics, Y-BC-JYLZJ was identified as *T. verruculosus.*

### 3.2. The Solubility of T. verruculosus to Insoluble Pi 

According to the results (Table 4), the content of Pi in the medium with *T. verruclosus* NBRIP increased obviously and the pH value decreased. The content of soluble Pi for the tricalcium phosphate and *T. verruclosus* treatment was more than 400.38% higher than that of the tricalcium phosphate treatment. The content of soluble Pi for the lecithin and *T. verruclosus* treatment was more than 832.36% higher than that of lecithin treatment. The hyphae Pi content for the tricalcium phosphate and *T. verruclosus* treatment was 13.58 mg·L^−1^, and the hyphae Pi content for the lecithin and *T. verruclosu* treatment was 12.41 mg·L^−1^. *T. verruculosus* reduced the pH of tricalcium phosphate and lecithin treated with *T. verruculosus* by 46.75% and 35.41%, respectively, compared with untreated tricalcium phosphate and lecithin.

### 3.3. Physical and Chemical Properties of Soil

The physical and chemical properties of the soil were tested during the pot experiment (data are the average of three repeated soil samples). The results show that the pH value of the soil was 6.67, the content of organic matter was 17.11 (mg·kg^−1^), the content of available nitrogen (N) was 8.02 (mg·kg^−1^), and the content of available phosphorus (P) was 17.05 (mg·kg^−1^). The available potassium (K) content was 52.15 (mg·kg^−1^).

### 3.4. Pi Content in Aboveground and Underground Parts

According to the results (Figure 3), the aboveground Pi content of cucumber seedlings treated with *T. verruculosus* was significantly higher than that of cucumber seedlings treated without *T. verruculosus.* The application of *T. verruculosus* under low phosphorus stress did not significantly improve the Pi content of plants, and the Pi content of the TPT treatment was 75.25% higher than that of the TP treatment. Under the treatment of LPT and TPT, the Pi content in the aboveground and underground parts was increased by 74.5% and 61.4%, respectively, indicating that *T. verruculosus* promoted the absorption of Pi in the aboveground parts. The results of the correlation analysis show that the surface morphological indexes, root parameters, chlorophyll content, chlorophyll a fluorescence parameters, and electron transport capacity of each RC were positively correlated (Figure 4).

### 3.5. Morphological Growth Index

The application of *T. verruculosus* significantly increased the growth indexes of cucumber seedlings (Table 5), and the aboveground growth indexes were positively correlated with the aboveground Pi content (Figure 4). Compared with the Pi treatment, the indexes of the DP treatment were significantly reduced, indicating that low Pi inhibited plant growth. The plant heights of PT, DPT, LPT, and TPT were 21.9%, 19.65%, 24.43%, and 21.3% higher than those of Pi, DP, LP, and TP, respectively. In terms of stem diameter, PT, DPT, LPT, and TPT were 16.4%, 8.1%, 10.2%, and 5.7% higher than Pi, DP, LP, and TP, respectively. The leaf areas of the LPT and TPT treatments were 86.7% and 62.13% higher than those of LP and TP, respectively. The aboveground dry weight and underground dry weight of plants treated with *T. verruculosus* were higher than those treated without *T. verruculosus*.

The application of *T. verruculosus* significantly increased the growth indexes of cucumber seedlings in the subsurface (Table 6). PT, DPT, LPT, and TPT treated with *T. verruculosus* had an increased root length, root surface area, root volume, mean diameter, and cross number as compared to those treated with no *T. verruculosus*. In the root scanning maps of each treatment (Figure 5A–H) and cross number analysis bar charts (Figure 5I–L), it can be clearly seen that the number of root tips and branches in each group treated with *T. verruculosus* was higher than in the group treated without *T. verruculosus*. Root morphological indexes increased with the increase in plant Pi content (Figure 4).

### 3.6. Effects of T. verruculosus Inoculation on Chlorophyll Content

According to the chlorophyll a content, chlorophyll b content, total chlorophyll content, and Chl a/Chl b ratio of the different treated plants (Table 7), the chlorophyll a contents of PT, DPT, LPT, and TPT were 50%, 24.71%, 35.21%, and 50% higher than those of Pi, DP, LP, and TP, respectively. Except PT and Pi, the chlorophyll b content of the other treatments had no significant difference, and the content of chlorophyll b in the Pi treatment increased by 42.47% when *T. verruculosus* was applied. The total chlorophyll contents of PT, DPT, LPT, and TPT were 77.24%, 33.62%, 153.7%, and 58.56% higher than those of Pi, DP, LP, and TP, respectively. The Chl a/Chl b ratios of the LPT and TPT treatments were 29.12% and 40.74% higher than those of the LP and TP treatments, respectively, while the Chl a/Chl b ratios of the other treatments were not significantly different. In combination with the correlation analysis, photosynthetic pigments were positively correlated with each morphological index and had a positive correlation with Fv/Fm, qP, ϕPSII, and ETo/RC for chlorophyll a fluorescence (Figure 4).

### 3.7. Chlorophyll a Fluorescence Parameters

Combined with the chlorophyll a fluorescence parameters (Figure 6A–D) and the fluorescence diagram results (Figure 6E), the Fv/Fm, ϕPSII, and qP in each group treated with *T. verruculosus* significantly increased, while the NPQ values decreased. The Fv/Fm, ϕPSII, and qP of the PT treatment were the highest, and the Fv/Fm, ϕPSII, and qP of the LPT treatment were higher than those of the TPT treatment. The Fv/Fm, ϕPSII, and the lowest qP were obtained for the DP treatment. Compared with PT, DPT, LPT, and TPT, the NPQ values of Pi, DP, LP, and TP were significantly increased. In addition to NPQ, the chlorophyll a fluorescence parameters increased with the increase in the aboveground Pi content (Figure 4).

According to the optical response curves of Y(II), qP, ETR, and NPQ under different treatments (Figure 7), it can be seen that, compared with the treatment without *T. verruculosus*, when the PAR value was less than 400 (µmol·m^−2^·s^−1^), the treatment with *T. verruculosus* significantly reduced the rate of Y(II) and qP reduction. When the PAR value was greater than 400 (µmol·m^−2^·s^−1^), Y(II) and qP under different treatments tended to be stable. Moreover, the degree of inhibition of light quantum production without the *T. verruculosus* treatment was greater than that under the *T. verruculosus* treatment. Compared with the treatment with *T. verruculosus*, the ETR growth rate was slower in the treatment without *T. verruculosus*. When the PAR value was greater than 400 (µmol·m^−2^·s^−1^), the NPQ of each group tended to be stable, and the NPQ of the treatment without *T. verruculosus* was higher than that of the treatment with *T. verruculosus*. The application of *T. verruculosus* can increase the Pi content of the plants, thus reducing the dissipation of excitation energy in the form of heat energy to a certain extent and alleviating PSII damage.

### 3.8. OJIP Curve Analysis

Period J was higher for the LPT treatment than in the other treatments. The PF intensity and I-P amplitude of the group treated with *T. verruculosus* in stage I and stage P were higher than those of the group treated without *T. verruculosus*, and the PT treatment had the highest intensity in stage I and stage P. The intensity of DP treatment was lowest in stage I and stage P (Figure 8A). Through the normalization of the fluorescence data, W_O-P_ was evaluated to further determine the sites where various electron transport chains were processed at the PSII receptor terminals. The normalized curve between the O and p steps was expressed as W_O-P_ = [(F_t_ − F_o_)/(F_m_ − F_o_)] (Figure 8B). In the O-J and J-I phases, the intensity of PF increased significantly under Pi stress, but there was no significant change in the intensity of PF in each treatment during the I-P phase. Through the normalization of the relative fluorescence between the O phase and the J phase (Figure 8C), it was found that no K value appeared at 300 s, indicating that the damage on the donor side was less than that on the recipient side.

As shown in Figure 9A–D, the ABS/RC, TR_o_/RC, ET_o_/RC, and DI_o_/RC of the cucumber seedling leaves without the *T. verruculosus* treatment were significantly higher than those treated with *T. verruculosus*. In Figure 9F, with the Pi treatment group as the control, except for the DPT treatment, φE_o_, Ψ_o_, and F_m_ were significantly higher for the PT, LPT, and TPT treatments than for the Pi treatment. This indicates that more light energy captured by PSII in plants treated with *T. verruculosus* is transferred to other electron acceptors downstream of QA. In addition, the capacity of the receptor pool is increased, and the number of active reaction centers per unit area is increased, which promotes electron transfer on the acceptor side of PSII. In Figure 9E, the ABS/RC, TR_o_/RC, and DI_o_/RC values of the DP, LP, and TP treatments were significantly higher than those of the control group, while the values of φE_o_ and Ψ_o_ were significantly lower. This indicates that, under Pi stress, the light energy absorbed by the PSII unit reaction center, the energy captured for reducing QA, and the energy dissipation of the PSII unit reaction center are relatively high, while the activity of the PSII reaction center, the acceptor capacity pool, and the electron transport rate are relatively low, and the number of active reaction centers per unit reaction center is relatively small. The increase in DI_o_/RC and ET_o_/RC indicates that insufficient Pi content leads to a decrease in the number of active reaction centers per unit area and a decrease in the maximum photochemical quantum yield, which causes certain damage to PSII electron transport, increases the burden of active reaction centers, and forces the increase in energy dissipation efficiency and heat dissipation [40], which, in turn, is also a reflection of the selfprotection mechanism. Therefore, the application of *T. verruculosus* can promote the absorption of Pi in plants, thereby improving the photochemical activity of plants and enhancing the light energy utilization efficiency.

## 4. Discussion

Since the discovery of the role of microorganisms in dissolving insoluble phosphorus in soil, researchers have unearthed many fungi with the ability to dissolve phosphorus. These include *Talaromyces flavus* [41], *Talaromyces funiculosus* [42], and *Talaromyces pinophilus* [43]. These strains, which belong to *Talaromyces* spp, have been reported to have phosphorus-dissolving abilities. In this study, the strain *Talaromyces* spp. was isolated from the maple. *T. verruculosus* and evaluated for its phosphorus-dissolving ability through a phosphorus-dissolving experiment. The results show that *T. verruculosus* had a good dissolving effect on tricalcium phosphate and lecithin. The pH value decreased as the concentration of available phosphorus in the culture medium increased. This finding is consistent with the results of Hwangbo’s study [44], which found a negative correlation between pH and available phosphorus content [45].

It has been proved that inoculation with PSM can improve the Pi absorption capacity of plants and promote plant growth [46,47,48]. The application of *T. verruculosus* was shown to improve the absorption capacity of TCP and lecithin in cucumber seedlings. Moreover, the plant height, leaf area, and root morphology of cucumber seedlings were higher than those without the application of *T. verruculosus*. The application of *Talaromyces flavus* can increase the plant height and dry biomass of rice, which is the same as the results of this study [49]. Water and nutrients in soil can be absorbed by plant roots and stably immobilize plants in the soil [50]. *T. verruculosus* can promote plant root growth to varying degrees, which helps plant roots to obtain more nutrients from the soil.

Photosynthesis plays an important role in plant metabolism [51,52]. The light energy absorbed by chlorophyll in photosynthesis is mainly used for photochemical reactions, heat dissipation, and emission in the form of fluorescence. The sum of quantum yields of these three pathways is 1. Since only the first two processes can be regulated, the intensity of fluorescence depends on the rate of photosynthesis and the rate of heat dissipation [53,54]. Therefore, measuring chlorophyll fluorescence parameters and rapid chlorophyll fluorescence induction kinetics well reflects the energy conversion efficiency and photosynthetic performance of plants. The results of this study show that the chlorophyll content was increased under the condition of inoculation with *T. verruculosus*, and the chlorophyll content played a certain role in regulating the chlorophyll fluorescence parameters. Under the condition of the same PAR value, the Y(II) value, qP, and ETR of the leaves increased with the increase in chlorophyll content. *T. verruculosus* could improve the ability of chloroplast to capture excitation energy. It has been reported that *Trichoderma* can increase chlorophyll synthesis in the leaves of host plants, thus enhancing the photosynthesis of plants and promoting the synthesis of plant organic matter [55]; however, the mechanism is still unclear. Therefore, the mechanism of promoting growth and the photosynthetic capacity of cucumber seedlings should be further explored.

Chlorophyll content is easily changed under stress, and photosynthesis was shown to be positively correlated with chlorophyll content [51]. Studies have shown that maize plants under low nitrogen stress can reduce the amount of nitrogen allocated to chlorophyll and photocapture proteins, thus reducing electron production and electron transport activity [56]. In this study, the chlorophyll content of leaves under low Pi stress decreased, the OJIP curve was flatter, φEo and ψo were lower, and the electron transport efficiency was correspondingly reduced. The application of *T. verruculosus* can increase the chlorophyll content and promote electron transport activity. When *T. verruculosus* was not applied, the DI_o_/RC and ET_o_/RC of the leaves increased and F_m_ decreased, resulting in a decrease in the number of active reaction centers per unit area and a decrease in the maximum photochemical quantum yield, which caused certain damage to the electron transfer of PSII. This is consistent with the results that showed that salt stress decreased the photosynthetic electron transfer capacity of the PSII donor side and the photosynthetic capacity of leaves. A study on the rapid fluorescence information of rapeseed under nutrient stress showed that nutrient loss resulted in a decreased electron transport quantum yield of PSII, a decreased energy capture efficiency of the PSII open reaction center, and caused PSII photoinhibition damage [57]. This is consistent with the results of low Pi stress in this study. The use of *T. verruculosus* can effectively alleviate this damage. Therefore, under low PI stress, *T. verruculosus* plays an important role in improving plant photosynthetic capacity and promoting plant growth.

According to the results of this study, *T. verruculosus* can enhance the availability of phosphorus in soil, alleviate the damage to the photosynthetic system caused by the external environment, and promote the growth of crops. It plays an important role in alleviating the lack of phosphorus resources, reducing the environmental pollution caused by the application of chemical phosphate fertilizers, and promoting sustainable agriculture.

## 5. Conclusions

In this study, 10 endophytic fungi with different characteristics were isolated from *Acer truncatum* stem segments, among which *T. verruculosus* had phosphorus-solubilizing potential. *T. verruculosus* can not only dissolve insoluble inorganic phosphorus, but also dissolve insoluble organophosphorus. The results show that *T. verruculosus* can expand phosphorus-soluble microbial resources. The pot experiment proved that *T. verruculosus* could regulate the phosphorus absorption and utilization capacity of cucumber seedlings under different phosphorus levels, promote the growth of cucumber seedlings, promote root elongation and branching, and expand the absorption range and capacity of seedling roots. Low Pi stress can inhibit plant photosynthesis and cause damage to the photosynthetic system. The application of *T. verruculosus* can regulate plant photosynthesis to a certain extent to alleviate the effects of low phosphorus stress on the photosynthetic system of cucumber seedlings, thus promoting their growth and development. This study provides a theoretical basis for the development and application of *T. verruculosus* in facility agriculture.

## Figures and Tables

**Figure 1 jof-10-00136-f001:**
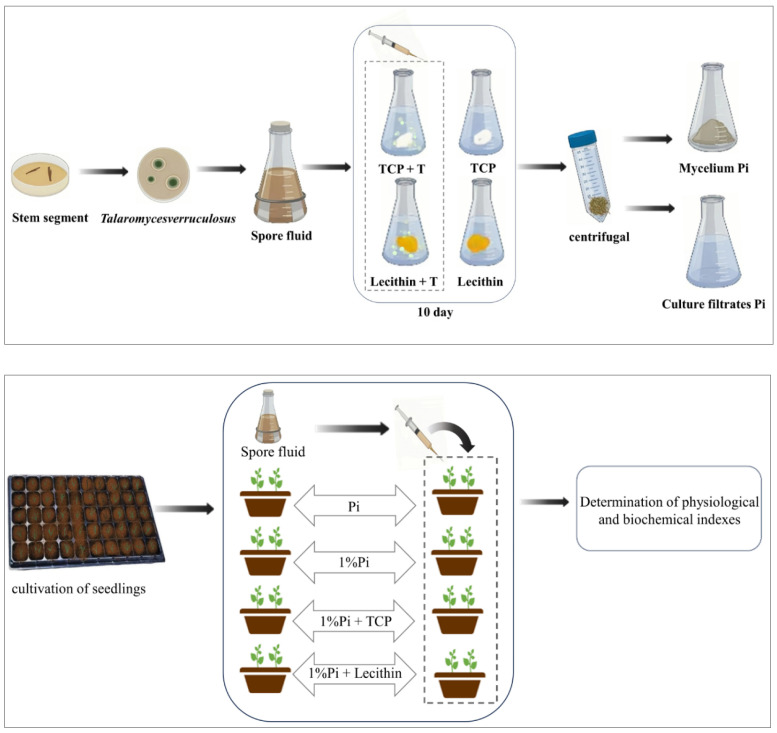
Experimental design flow chart.

**Figure 2 jof-10-00136-f002:**
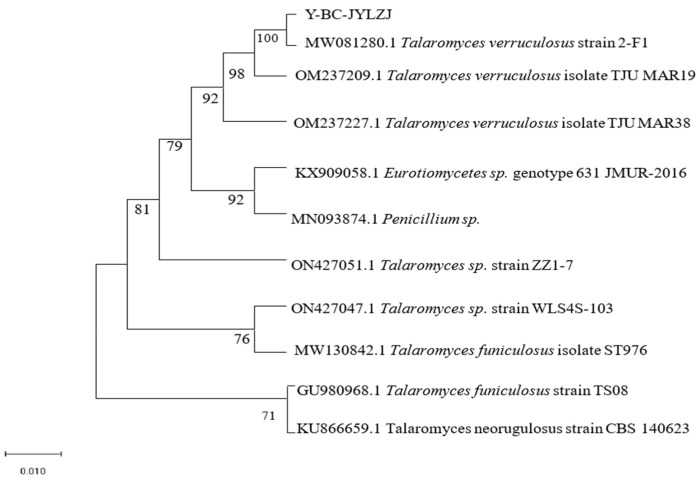
Phylogenetic tree of Y-BC-JYLZJ.

**Figure 3 jof-10-00136-f003:**
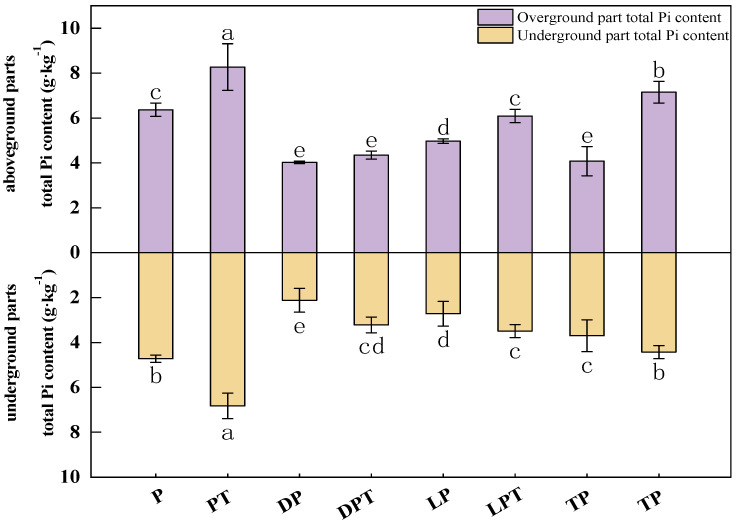
Pi contents in aboveground and underground parts of cucumber seedlings. Values are expressed as means ± SD (*n* = 6). Different letters denote significant differences between treatments (*p* < 0.05).

**Figure 4 jof-10-00136-f004:**
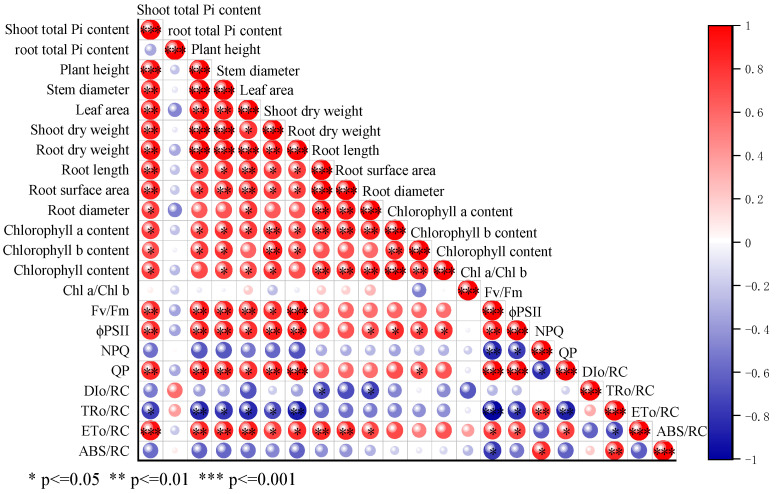
Correlation analysis of cucumber seedling morphology and light sum index.

**Figure 5 jof-10-00136-f005:**
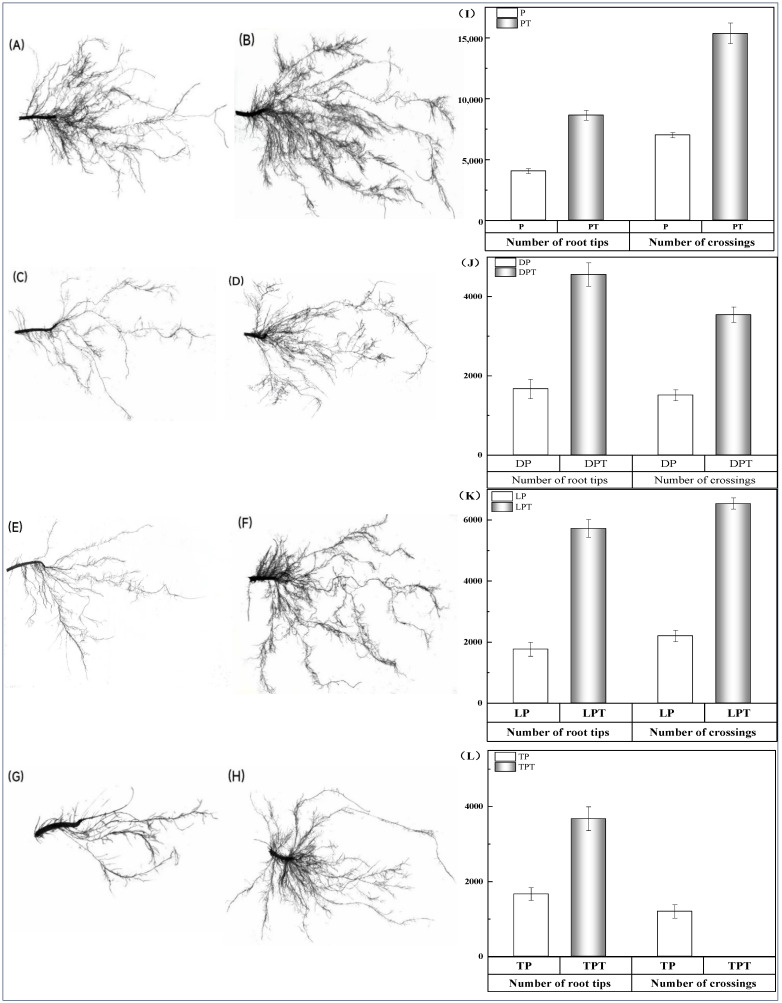
Comparison of root scan of cucumbers under different treatments, number of cross roots, and number of root tips. (**A**–**H**) root scan map. (**A**) Pi; (**B**) PT; (**C**) 1% Pi; (**D**) DPT; (**E**) LP; (**F**) LPT; (**G**) TP; (**H**) TPT; (**I**–**L**) comparison between the number of root tips and the number of crossings. (**I**) P, PT; (**J**) DP, DPT; (**K**) LP, LPT; (**L**) TP, TPT.

**Figure 6 jof-10-00136-f006:**
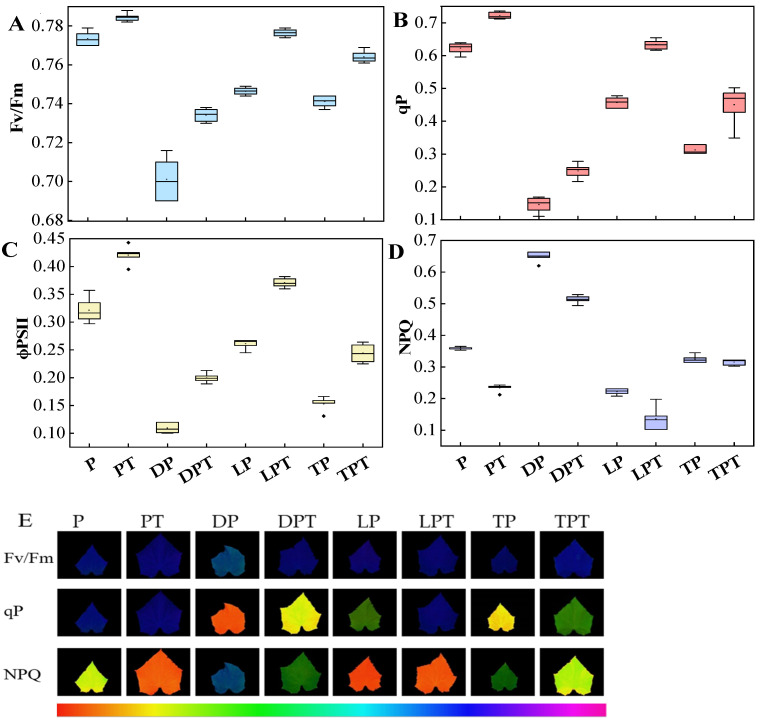
Effects of different treatments on chlorophyll fluorescence kinetics of photosystem II (PS II) in cucumber: (**A**) maximum quantum yield of PSII (Fv/Fm); (**B**) coefficient of photochemical quenching (qP); (**C**) effective quantum yield of PSII (ϕPSII); (**D**) nonphotochemical quenching (NPQ); (**E**) photosystem II (PS II) chlorophyll fluorescence images of cucumber plants under different treatments. Values are expressed as means ± SD (*n* = 6).

**Figure 7 jof-10-00136-f007:**
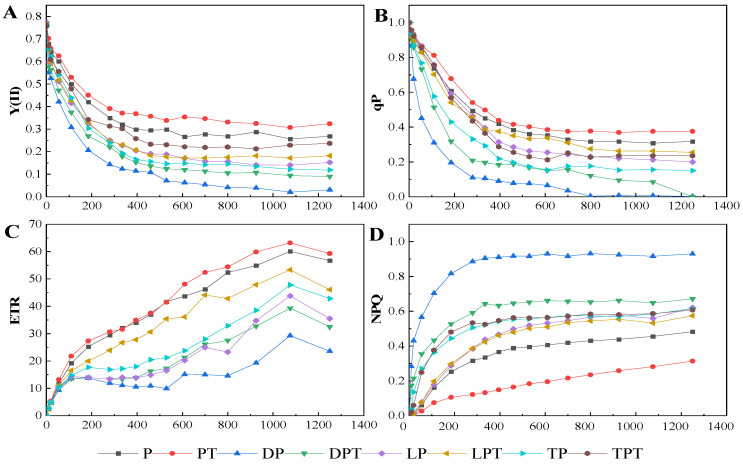
Function relationship between Y (II) (**A**), qP (**B**), ETR (**C**), nonphotochemical quenching (NPQ) (**D**), and PAR of cucumber leaves under different treatments. Values are expressed as means ± SD (*n* = 6).

**Figure 8 jof-10-00136-f008:**
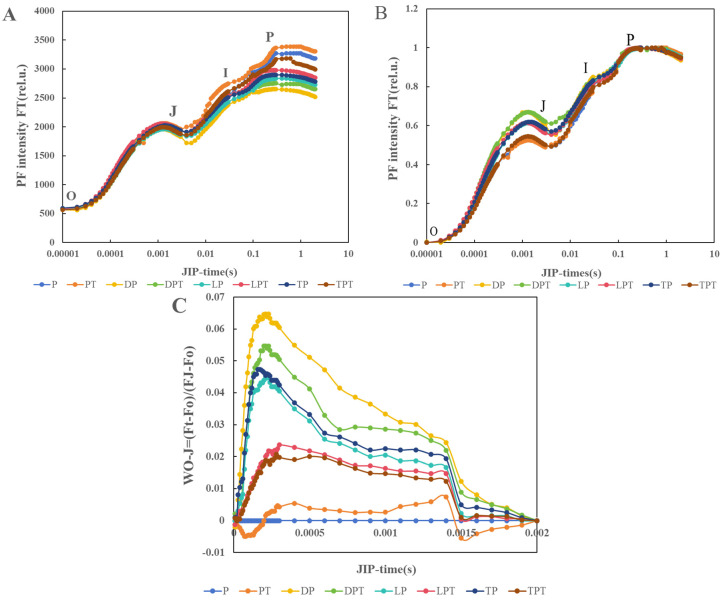
The predictive fluorescence (PF) curve of cucumber leaves (**A**), and the normalization curve between O and P orders is expressed as W_O-P_ = [(F_t_ − F_o_)/(F_m_ − F_o_)] (**B**). The normalization between order O and order J suggests that the fluorescence (PF) curve is expressed as W_O-J_ = [(F_t_ − F_o_)/(F_J_ − F_o_)] (**C**) with an average value of ±SD (*n* = 6).

**Figure 9 jof-10-00136-f009:**
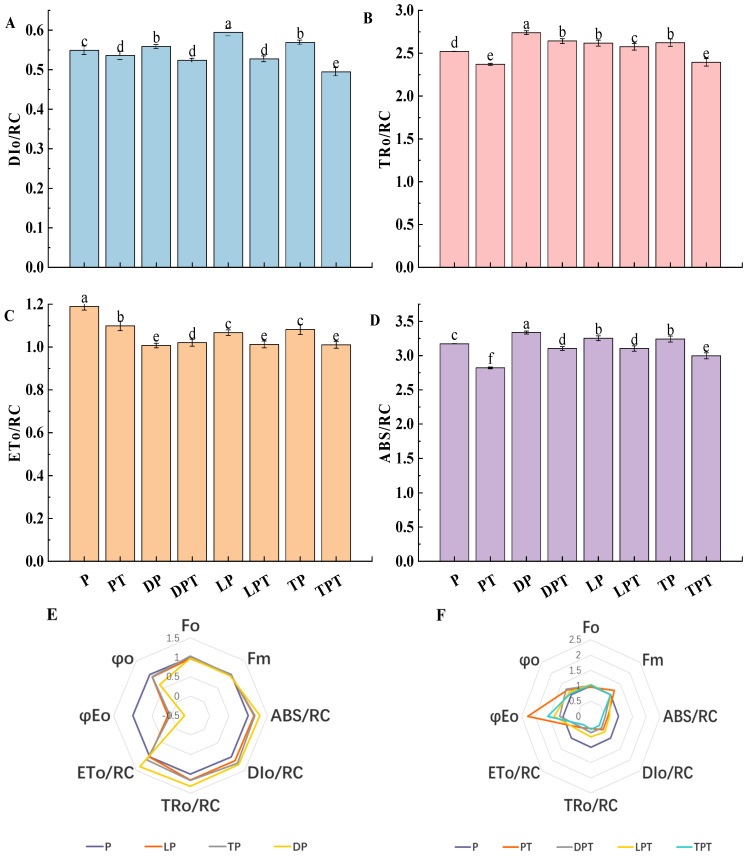
Dissipated energy flux (t = F_o_) per reaction center (DI_o_/RC) (**A**), captured energy flux per reaction center (t = F_o_) (TR_o_/RC) (**B**), electron transport flux per reaction center (t = F_o_) (ET_o_/RC) of cucumber leaves under different treatments (**C**), and absorbed flux per reaction center (RC) (ABS/RC) (**D**). Radar map of the changes in JIP parameters in the leaves of cucumber seedlings treated without *T. verruculosus* (**E**). Radar map of the changes in JIP parameters in the leaves of cucumber seedlings treated with *T. verruculosus* (**F**). Values are expressed as average ± SD (*n* = 6). Different letters denote significant differences between treatments (*p* < 0.05).

**Table 1 jof-10-00136-t001:** Chlorophyll fluorescence parameters and calculation equations.

Chlorophyll Fluorescence Parameter	Formula
Maximum efficiency of PSII photochemistry (Fv/Fm)	(Fm − Fo)/Fm
photochemical quenching (qP)	(Fm′ − Fs)/(Fm′ − Fo′)
nonphotochemical quenching (NPQ)	(Fm − Fm′)/Fm′
electron transport rate (ETR)	PAR × ΦpsII × 0.84 × 0.5

**Table 2 jof-10-00136-t002:** JIP−test parameters derived from fast chlorophyll a fluorescence kinetics OJIP.

Parameters	Explanation
F_o_	Minimum fluorescence
F_m_	Maximum fluorescence
F_j_	2 ms instantaneous fluorescence
V_j_ = (F_j_ − F_o_)/(Fm − F_o_)	J-phase relatively variable fluorescence
Ψ_o_ = (1 − Vj)	The captured light energy was used for the quantum yield of QA—downstream electron transport
φE_o_ = [1 − (F_o_/F_m_)](1 − V_j_)	The absorbed energy was used for the quantum yield of electron transport
ABS/RC	Energy absorbed per unit reaction center
TR_o_/RC	Energy captured per unit reaction center
ET_o_/RC	Energy transferred by electrons per unit reaction center
DI_o_/RC	Dissipated energy flux per RC

**Table 3 jof-10-00136-t003:** Results of Blast sequence comparison for *Acer truncatum* endophytic bacteria.

Organism	Sequencing Organism Number
*Alternaria compacta*	1.9.15
*Trichoderma atroviride*	7.21
*Coprinellus radians*	2.18
*Alternaria porri*	3.5.6
*Actinomucor elegans*	4.10.11
*Chaetomium globosum*	8.12.17
*Talaromyces funiculosus*	13
*Trichoderma asperellum*	16
*Talaromyces verruculosus*	19
*Alternaria tenuissima*	20
No similar sequences were found in Blast	14

**Table 4 jof-10-00136-t004:** Pi content and pH value of culture medium of filtrate and hyphae under different conditions.

Treatment	Pi Content in Culture Medium (mg·L^−1^)	Hyphae Pi Content (mg·kg^−1^)	pH
TCP + Tv	105.63 ± 1.58 b	13.58 ± 0.04 a	3.85 ± 0.23 b
TCP	21.11 ± 2.79 c	0.00 ± 0.00 c	7.23 ± 0.12 a
PC + Tv	268.80 ± 22.46 a	12.41 ± 0.33 b	2.28 ± 0.12 c
PC	28.83 ± 8.31 c	0.00 ± 0.00 c	3.53 ± 0.42 b

Note: Values are expressed as means ± SD (*n* = 6). Different letters denote significant differences between treatments (*p* < 0.05). TCP—tricalcium phosphate; PC—lecithin; Tv—*T. verruculosus.*

**Table 5 jof-10-00136-t005:** Effects of *T. verruculosus* on morphological indexes of cucumber. Note: Values are expressed as means ± SD (*n* = 6). Different letters denote significant differences between treatments (*p* < 0.05).

Treatment	Height of Plant (cm)	Stem Thick (mm)	Area of Leaf (cm^2^)	Above Ground Dry Weight (g)	Underground Dry Weight (g)
P	29.88 ± 1.92 b	5.07 ± 0.38 b	100.96 ± 3.28 b	1.90 ± 0.12 b	0.36 ± 0.05 b
PT	36.42 ± 10.67 a	5.9 ± 0.11 a	120.39 ± 6.62 a	2.60 ± 0.17 a	0.42 ± 0.03 a
DP	16.9 ± 2 e	4.22 ± 0.37 c	45.05 ± 7.46 f	1.20 ± 0.31 d	0.14 ± 0.03 e
DPT	20.22 ± 1.05 de	4.56 ± 0.34 bc	74.36 ± 5.91 c	1.38 ± 0.28 cd	0.2 ± 0.02 d
LP	21.98 ± 0.77 cde	4.61 ± 0.32 bc	53.83 ± 8.16 e	1.60 ± 0.31 bcd	0.23 ± 0.04 d
LPT	27.35 ± 1.34 bc	5.08 ± 0.46 b	100.48 ± 6.16 b	1.75 ± 0.32 bc	0.32 ± 0.03 bc
TP	23.23 ± 1.35 cd	4.77 ± 0.26 b	66.2 ± 4.6 d	1.40 ± 0.32 cd	0.23 ± 0.03 d
TPT	28.18 ± 1.53 bc	5.04 ± 0.32 b	107.33 ± 7.95 b	1.63 ± 0.29 bcd	0.31 ± 0.03 c

**Table 6 jof-10-00136-t006:** Effect of *T. verruculosus* on morphological indexes of cucumber.

	Root Length (cm)	Root Surface Area (cm^2^)	Root Volume (cm^3^)	Root Diameter (mm)	Number of Forks
P	1200.92 ± 457.44 cd	119.7 ± 46.7 cd	0.95 ± 0.4 cd	0.32 ± 0.02 ab	1772.5 ± 680.79 cde
PT	2930.58 ± 756.43 a	316.77 ± 89.76 a	2.73 ± 0.85 a	0.34 ± 0.02 a	5180.33 ± 1478.36 a
DP	573.38 ± 135.4 d	56.32 ± 16.46 d	0.44 ± 0.15 d	0.31 ± 0.02 ab	766.17 ± 327.6 de
DPT	1509.74 ± 393.12 c	158.52 ± 49.12 bc	1.33 ± 0.48 bc	0.33 ± 0.02 ab	2423.67 ± 892.86 dc
TP	505.5 ± 371.77 d	49.6 ± 36.01 d	0.39 ± 0.28 d	0.31 ± 0.02 ab	579 ± 545.34 e
TPT	1397.45 ± 508.02 c	144.87 ± 59.55 c	1.2 ± 0.55 bcd	0.33 ± 0.02 ab	2170.83 ± 1049.51 bcd
LP	586.61 ± 174.15 d	56.01 ± 19.16 d	0.43 ± 0.18 d	0.3 ± 0.03 b	743.5 ± 283.29 de
LPT	2145.19 ± 649.09 b	221.18 ± 71.08 b	1.82 ± 0.62 b	0.33 ± 0.02 ab	3448.5 ± 1528.79 b

Note: Values are expressed as means ± SD (*n* = 6). Different letters denote significant differences between treatments (*p* < 0.05).

**Table 7 jof-10-00136-t007:** Effects of Pi and *T. verruculosus* on chlorophyll content of cucumber leaves.

Treatment	Chlorophyll a Content (mg/g)	Chlorophyll b Content (mg/g)	Chlorophyll Content (mg/g)	Chl a/Chl b
Pi	1.02 ± 0.1 c	0.73 ± 0.04 b	1.45 ± 0.13 c	1.39 ± 0.18 b
PT	1.53 ± 0.19 a	1.04 ± 0.36 a	2.57 ± 0.39 a	1.62 ± 0.32 b
DP	0.85 ± 0.05 cd	0.48 ± 0.07 bc	1.16 ± 0.11 d	1.77 ± 0.39 b
DPT	1.06 ± 0.08 bc	0.49 ± 0.06 b	1.55 ± 0.09 bc	1.89 ± 0.3 b
TP	0.8 ± 0.09 cd	0.49 ± 0.04 bc	1.11 ± 0.12 d	1.62 ± 0.23 b
TPT	1.2 ± 0.14 b	0.55 ± 0.16 c	1.76 ± 0.27 b	2.28 ± 0.42 a
LP	0.71 ± 0.11 d	0.39 ± 0.04 d	0.54 ± 0.15 e	1.82 ± 0.3 b
LPT	0.96 ± 0.08 bcd	0.41 ± 0.04 c	1.37 ± 0.1 cd	2.35 ± 0.23 a

Note: Values are expressed as means ± SD (*n* = 6). Different letters denote significant differences between treatments (*p* < 0.05).

## Data Availability

The data used in this study may be obtained from the corresponding author upon submission of a reasonable request.

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
