# Peer review of "Isolation and Identification of Acer truncatum Endophytic Fungus Talaromyces verruculosus and Evaluation of Its Effects on Insoluble Phosphorus Absorption Capacity and Growth of Cucumber Seedlings"

_jof, 2024, doi:10.3390/jof10020136_

Round 1
Reviewer 1 Report
Comments and Suggestions for Authors
The authors conducted an interesting study with significant prospects. However, it is necessary to improve the manuscript so that it becomes more ready for publication.
1. The introduction should be carefully corrected. Some sentences are broken off, and some phrases are repeated from sentence to sentence. It is necessary to state very clearly and consistently the role of phosphorus in the biochemical processes of plants - not at the level of "it affects plant growth", but more specifically. Moreover, the relationship between phosphorus and chlorophylls must be substantiated in order to understand why the authors paid so much attention to this in the experimental part.
2. It is necessary to describe the location of the plant more clearly. Specifying the exact coordinates and a more accurate reference to the terrain is welcome. It is necessary to clearly describe from which part of the plant the sample was taken to isolate the fungi, whether it was only one plant and only one sample. If only one sample from one plant was used for fungal isolation, this, of course, reduces the significance of the results obtained in the sense that it cannot be unequivocally stated that it is this endophyte that provides Acer truncatum environmental resistance.
3. The authors do not give any explanation why exactly one strain was chosen for identification and research. The description "21 strains have been isolated" is very poor. Table 3 looks incorrect. The logic of the study indicates that all isolated strains had to be tested for their ability to influence plant growth, and the most active strain could be studied in detail. However, the authors don't write anything about it, and it needs to be corrected.
Conclusions "In this study, 10 endophytic fungi with different characteristics were isolated from Acer truncatum stem segments, among which Talaromyces verruculosus had phosphorus solubilizing potential." confirms my assumption that the authors need to check everything carefully.
4. The authors chose very cumbersome designations of processing options in the study: Pi (1mM KH2PO4), Pi) ,Pi+ Talaromycesverruculosus (Pi+T), apply low concentration Pi (10uM 116 KH2PO4), 1%Pi) ,1%Pi+ T,1%Pi+Ca3(PO4)2 (TPi) ,1%Pi+ Ca3(PO4)2 +T (T%Pi + T) ,1%Pi+ Lec- 117 ithin (LP),1%Pi+ lecithin +T(LP+T).
I think it needs to be replaced with a more readable one, such as A, B, C, etc.
5. The authors write that "The experimental treatment was started at the four-leaf single-phase stage, and the spore solution was diluted to 1×108cfu/mL, then the root was irrigated once a week for a total of 4 weeks."
Is it a question here that the spores of the fungus continued to be introduced once a week? Or was the watering carried out with water? If the irrigation was carried out with water, which one exactly? It is also necessary to accurately describe the growing regime of cucumber seedlings: temperature, humidity, lighting, which soil was used, whether an analysis of the soil composition was carried out confirming uniformity.
Table 5 named "Physicochemical properties of soil used before and after the experiment." However, this table contains such columns as "Height of plant (cm)", "Stem thick (mm)", "Area of leaf (cm2)" and others. Please check the contents.
6. The discussion should be verified. For example, the sentences " researchers have unearthed many archaea, bacteria, fungi and algae with the ability to dissolve phosphorus. These include Talaromyces flavus[37], Talaromyces funiculosus[38] and Talaromyces pinophilus[39]" (Lines 425-427). Only fungi are listed here, but the authors mention other organisms first.
7. Links to tables and figures should be removed from the Discussion. There is no need to describe the results again in the Discussion. It is necessary to compare your data with the available literature and justify exactly how the fungus can have the observed effect. The authors point out that several Thalaromyces strains have an effect on plant growth. I would like to discuss why this is happening and what the mechanisms are. Do Penicillium and Aspergillus strains have similar effects?
8. So, I repeat that the conclusions should be brought into line with the stated results.
Comments on the Quality of English Language
The text contains typos, and some sentences look unfinished. I am not a native speaker, but I believe that English should be corrected.
Reviewer 2 Report
Comments and Suggestions for Authors
Comments to the Authors
Authors should consider the formatting of the Article genus and species of space, uppercase letters, and italicization.
You can stick to a specific format for T. verruculosus except for the initial indication. There are multiple instances of repetition throughout the paper.
What is the relationship between plants' chlorophyll content and electron transport activity when T. verrucous is treated? Authors should provide a comprehensive analysis accompanied by credible citations.
The theoretical basis of evidence needs to substantiate the correlation between the formers and agricultural productivity. Authors should revise the concluding sentence using appropriate scientific terms.
Round 2
Reviewer 1 Report
Comments and Suggestions for Authors
The authors significantly changed the manuscript and it became suitable for publication. However, I recommend checking the text carefully, especially the Latin names of plants and fungi.
